



# Density distribution across the Alpine lithosphere constrained by 3D gravity modelling and relation to seismicity and deformation

Cameron Spooner [1,2], Magdalena Scheck-Wenderoth[1,3], Hans-Jürgen Götze[4], Jörg Ebbing[4], György Hetényi[5]

[1]GFZ German Research Centre for Geosciences, Potsdam, Germany
[2]Institute of Earth and Environmental Science, Potsdam University, Potsdam, Germany
[3]Department of Geology, Geochemistry of Petroleum and Coal, RWTH Aachen University, Aachen, Germany
[4]Institute of Geosciences, Christian-Albrechts-University Kiel, Kiel, Germany
[5]Institute of Earth Sciences, University of Lausanne, Lausanne, Switzerland

*Correspondence to*: Cameron Spooner (spooner@gfz-potsdam.de)

**Abstract.** The Alpine Orogen formed as a result of the collision between the Adriatic and European plates. Significant crustal heterogeneity exists within the region due to the long history of interplay between these plates, other continental and oceanic blocks in the region, and inherited crustal features from earlier orogenys. Deformation relating to the collision continues to the present day. Here, a seismically constrained, 3D, structural and density model of the lithosphere of the Alps and their respective
forelands, derived from integrating numerous geoscientific datasets, was adjusted to match the observed gravity field. It is shown that the distribution of seismicity and deformation within the region correlates strongly to thickness and density changes within the crust, and that the present day Adriatic crust is both thinner and denser (22.5 km, 2800 kg/m3) than the European crust (27.5km, 2750 kg/m3). Alpine crust derived from each respective plate is found to show the same trend with zones of Adriatic provenance (Austro-Alpine and Southern Alps) found to be denser and those of European provenance (Helvetic Zone
and Tauern Window) to be less dense suggesting the respective plates and related terrains had similar crustal properties to the present day prior to orogenesis. The model generated here is available for open access use to further discussions about the crust within the region.

## 1 Introduction

The Alps are one of the best studied mountain ranges in the world, yet significant unknowns remain regarding their crustal
structure and any links that may exist between the localisation of deformation and seismicity in the region and crustal heterogeneity. Significant amounts of seismicity and deformation within the region correspond to plate dynamics, such as at the convergence of the European and Adriatic plates in North-East Italy (Restivo et al., 2016) where the Adriatic plate is observed to act as a rigid indenter, moving northwards and rotating counter-clockwise into the weaker European plate (Nocquet and Calais, 2004; Vrabec and Fodor, 2006). However, numerous large historic seismic events (Fäh et al., 2011; Stucchi et al.,
2012; Grünthal et al., 2013), such as the Magnitude 6.6 Basel earthquake in 1356 AD, lie substantially intra-plate in areas with



low amounts of horizontal surface strain (Sánchez et al., 2018) suggesting that features within the crust are also significant factors to their localisation.

Crustal heterogeneities on the European plate, constituting the northern foreland of the Alps, principally derive from different terranes that collided during the Carboniferous age Variscan orogeny (Franke, 2000). Collision during orogenesis resulted in

the juxtaposition of crustal domains with differing properties next to one another, such as Moldanubia and Saxothuringia, (Babuška and Plomerová, 1992; Freymark at al., 2016) and also resulted in the creation of the Vosges, Black Forest and Bohemian massifs.

As a consequence of the collision of the Adriatic plate with the European plate from the Cretaceous until the present (Handy et al., 2010), heterogeneity within the Alpine orogen is also very pronounced, however differing interpretations exist on the

plate provenance of some features. Traditionally, the Alps have been split into distinct zones according to their plate of origin and metamorphic history, such as the Adriatic derived Austro-Alpine and Southern Alps, the European derived Helvetic Alps and the Penninic zone representing distal margin units and slivers of oceanic crust (Schmid et al., 1989). The Brianconnais crustal block that lies within the Penninic zone derives from the Iberian plate (Frisch, 1979). Newer works examining the plate provenance of Alpine zones have reinterpreted some features such as the Tauern Window from Penninic origin to European

plate origin (Schmid et al., 2004).

Density distribution throughout the crust of the region is also affected by mantle features and regions of thinned crust resulting in sedimentary depocentres. The crust is thinned at three main depocentres within the region, the Po Basin of the southern foreland, the Molasse Basin of the northern foreland and the Upper Rhine Graben, also within the northern foreland, that formed as part of the European Cenozoic Rift System in the Eocene (Dèzes at el., 2004). Anomalously high densities within

the crust are present in the Western Alps, within the Ivrea Zone, as a result of a South East dipping mantle wedge, where mantle and lower crustal rocks are present at upper crustal depths (Zingg et al., 1990) and even at the surface (Pistone et al., 2017 and therein).

Previous published interpretations of crustal features within the orogen have been primarily based upon 2D seismic sections (e.g Brückl et al., 2007), tending to result in simplistic models. However, significant lateral differences in crustal structure

have been demonstrated, through the deployment of parallel seismological profiles, spaced 15 km apart in the Eastern Alps (Hetényi et al. 2018), indicating the need for more complex models. Studies that have integrated multiple geo-scientific datasets to create 3D models of the region, have either focussed on smaller sub-sections of the Alps (Ebbing, 2002) or included the Alps as part of a much larger study area (E.g. Tesauro et al., 2008). Therefore the generation of a 3D, crustal scale, gravity constrained, structural model of the Alps and their forelands at an appropriate resolution could be used to more accurately

describe crustal heterogeneity in the region. The generation of such an Alpine-wide specific model is made possible by the existence of seismological results from numerous published deep seismic surveys that have been completed throughout the region and available high quality global gravity field models. Within this current work, such data is integrated to give insights into the distribution of densities within the crust as constrained by 3D gravity modelling across the vast majority of the Alpine



region and its forelands for the first time, so that questions about the relationship between the distribution of densities within

the crust and seismicity and deformation patterns can be answered.

## 2 Input Data

Existing geological and geophysical observations from previous published works of the Alps and their respective forelands were used as constraints for the generation of the 3D structural model. Topography and bathymetry were utilised unaltered from ETOPO1 (Amante and Eakins, 2009), as show in Fig. 1a. The data integrated for the constraint of sub-surface lithospheric

features is shown in Fig. 1b and includes: regional scale, gravitationally and seismically constrained models of the TRANSALP study area (Ebbing, 2002), the Molasse Basin (Przybycin et al., 2014) and the Upper Rhine Graben (Freymark et al., 2017); regional scale, seismically constrained models of the Po Basin, such as MAMBo (Turrini at al., 2014; Molinari et al., 2015); and seismic reflection heights and their associated P wave velocity from projects such as ALP'75, EGT'86, TRANSALP, ALP 2002 and EASI (IESG, 1978; IESG & ETH Zuerich, 1981; Strößenreuther 1982; Mechie et al 1983; Zucca 1984; Gajewski &

Prodehl 1985; Deichmann et al, 1986; Gajewski et al 1987; Gajewski & Prodehl 1987; Yan and Mechie 1989; Zeis et al 1990; Aichroth et al 1992; Guterch et al 1994; Ye et al 1995; Scarascia and Cassinis, 1997; Enderle et al 1998; Bleibinhaus & Gebrande, 2006; Brueckl et al., 2007; Hetényi et al., 2018). The top surface of the asthenospheric mantle (LAB) was utilised unaltered from Geissler (2010).

Input data coverage for the constraint of most sub-surface lithospheric features was sufficient however thicknesses of

unconsolidated sediments were not available across the full modelled region. In regions  of less dense data coverage, continental scale, seismically constrained, integrative best fit models, EuCRUST-07 and EPcrust (Tesauro et al., 2008; Molinari and Morelli, 2011) were used in addition. Both models provided complete coverage of major structural interfaces and P wave velocities over the whole modelled area at a coarse resolution. Detailed values of unconsolidated sediment thicknesses were only available in the Upper Rhine Graben, the Molasse Basin and the Po Basin, as the seismic sections utilised

lacked the resolution for shallower features and the continental scale models did not differentiate between sedimentary strata. The free-air anomaly utilised was calculated from the global gravity model EIGEN-6C4 (Förste et al., 2014), at a fixed height of 6 km above the datum (Fig. 2, further referred to as observed gravity). As the gravity data source is a hybrid, terrestrial and satellite dataset, the potential exists for it to be lacking some of the short wavelength response that a fully terrestrial dataset would possess. The fixed height of 6 km was utilised to account for this, so that the vertical component of the gravity response

from the generated structural model (further referred to as calculated gravity) and observed gravity can be directly compared during the gravity modelling process.

## 3 Method

Data from numerous existing geoscientific datasets (see Input Data section) were integrated to create a gravity constrained, 3D, structural and density model of the lithosphere of the Alps and their respective forelands. The study area of this work,

indicated in both Figs. 1a and b, covers a region of 660 km x 620 km where the highest density of data source coverage was



available. The vast majority of the Alps and their forelands are included, with the Central and Eastern Alps and the Northern Foreland the best covered regions.

The software package Petrel (Schlumberger, 1998) was utilised for the creation and visualisation of the modelled surfaces in 3D, representing the key structural and density contrasts within the region. These surfaces were: 1. top water; 2. top
unconsolidated sediments; 3. top consolidated sediments; 4. top upper crust; 5. top lower crust; 6. top lithospheric mantle; 7. top asthenospheric mantle. All surfaces were generated with a grid resolution of 20 km x 20 km using Petrel's convergent interpolation algorithm.

Layers of the model were generated by correlation and integration between data sources, with the exception of the following: 1. The water layer was generated from cropping ETOPO1 to 0 m a.s.l. No freshwater bodies were added as they are too small
to be of impact at the model resolution utilised; 2. The top unconsolidated sediment surface used in the modelling corresponds to topography and bathymetry, which is plotted in Fig. 1a; 3. As a result of unconsolidated sediment thicknesses from the data sources only being present in the Upper Rhine Graben, the Molasse Basin and the Po Basin, outside of these regions a thickness of 0 was used. This was deemed acceptable due to the lack of unconsolidated sediment thicknesses, large enough to be of impact at the model resolution utilised, outside of these regions; 4. The LAB was utilised unedited from the data source as it
does not represent a significant density contrast and was therefore deemed sufficient to utilise a lower resolution existing model. Alpine nappe stacks were included within the consolidated sediments layer of the model.

During correlation and integration, a hierarchy of data source types was used and in the case of contradiction between the different data sources those of the highest hierarchy were taken. The hierarchy was derived from the quality, resolution and consistency of data sources and was as follows: 1. regional scale, gravitationally and seismically constrained models; 2.
regional scale, seismically constrained models; 3. individual seismic reflection surfaces and interpreted sections; 4. continental scale, seismically constrained, integrative best fit models.

No subduction interfaces were modelled, which was deemed acceptable as multiple studies within the region have shown that the effect of different subduction polarities as well as the presence or lack of subducting plates is small. Previous 2D gravity modelling work across the TRANSALP profile has demonstrated that the differences in gravity response between a model of
both different subduction polarities and a model setup with no subducted crust were marginal (Deutsch, 2014). Work into the contribution of subducting slabs in the region to the gravity field have also shown that their impact is also relatively small, in the region of 30 mGal (Lowe, 2019).

The 3D gravity modelling software used for constraining the generated structural model was IGMAS+ (Schmidt et al., 2010) which operates by creating triangulated meshes between points on input surfaces and vertical parallel planes, around a body
of homogenous density, to calculate their volumetric contribution to the gravity response. Gravity in the model was calculated at 6 km above the datum to be concurrent with the observed gravity. By doing so the short wavelength response of the calculated gravity was not overestimated as mentioned prior. The top of the model was also set to a height of 6 km with a density of 0 used to represent the column of air between it and topography. The model was set up with a plane spacing of 20 km and constant 20 km spaced points from the correlated input surfaces, resulting in a model grid of 20 km x 20 km horizontal





resolution. To account for the edge effect of the gravity field, the model was extended by 50% (330 km) in all directions of the studied area using the surfaces from EuCrust-07 (Tesauro et al., 2008).

The Free Air gravity response was utilised as this work is focussed on the crustal composition of the Alpine region and as up to 4.8km of crust lies above sea level within the modelled area, removing this from the gravity signal was deemed unacceptable. Additionally, the complex geological makeup of the Alps means that the removal of Alpine topography as a Bouguer slab of
homogenous density potentially introduces errors.

The process of gravity modelling utilised the alteration of an initial 3D structural model, comprising surface heights and densities, such that through multiple iterations the resulting model produced a similar gravity field to that of the observed gravity. Best practice of such an iterative process allows only one input parameter, density or surface heights, to be altered. Here, the surfaces generated as part of the integration work were used unaltered during the gravity modelling phase as they
were deemed to be better constrained than the densities from the input data, leaving only density as a free parameter.

For the calculation of the densities used in the initial structural model, P wave velocities from seismic data sources were converted using the experimentally derived empirical relationship detailed in Brocher (2005). In the absence of seismic data, P wave velocities from the continent scale models listed in the Data Sources section were used to supplement, giving coverage over the entire study area. The water layer was assumed to be a density of 1025 kg/m2 and the asthenospheric mantle layer
3320 kg/m2.

The densities derived from the P wave velocity conversions were then used in conjunction with densities from the input regional scale, gravitationally and seismically constrained models, to split the layers of the generated model laterally into domains of different density, to reflect the heterogeneous nature of the crust within the region. During the generation of the model, preference was given to resolving major densities contrasts. This involved cases of grouping units of known differing
lithology, age and/or provenance together, should they appear to have a similar density so as to best fit the gravity in the region. An overview of all the mean densities of each density body of the model, derived from the seismic P wave velocities, is found in Table 1.

To identify how well the structural model created fits the gravity field in the region, the calculated gravity was then taken away from observed gravity during interactive alterations of the location of different domains within each layer and their densities,
and the result (further referred to as residual anomaly) interrogated. No filtering for specific wavelengths was done during gravity modelling, with the full signal utilised at all times. No presumptions were made over which tectonic features would require domains of different density, with their location ultimately derived from the gravity modelling process.

In the case of anomalies in the residual gravity field, the depth of the source was estimated to be half the width of the anomaly wavelength and the density of the body lying at that depth was increased for a positive residual anomaly or decreased for a
negative residual anomaly. Successive iterations of the model were then generated by modifying the distribution of densities within the model layers. This was repeated until a 3D structural density model of the region was obtained, that best reproduces the indications of both the seismic data sources and the gravity field.



## 4 Results

Figure 3 shows a North-South cross section through the generated model illustrating the thickness of all main structural layers
of the model, the density domains defined within them and the calculated and observed gravity of the section. The location of
the cross section can be seen in Fig. 1a but is also marked on all figures illustrating the setup of the model.

Key features noted from the setup of the model indicate more heterogeneity is required in the crust, than in other model layers,
to replicate the gravity field and that significant differences exist between the crust of the European and Adriatic plates.
Sedimentary thicknesses, both unconsolidated and consolidated, are thinner in the Molasse Basin than in the Po Basin and
crustal densities and thicknesses can also be observed to differ between the plates. In the orogen itself, the result of
incorporating all Alpine nappes within the consolidated sediment layer can be observed, with higher thicknesses in the central
Alps. The crust is thickest below the central Alps and is compensated for by a higher thickness and density of the lower crust.
The observed gravity along with the calculated gravity of the model can also be observed, indicating a close fit.

Figure 4 shows the thicknesses of the layers of the generated model that were created as a result of the correlation and
integration work, with the areal extent of all utilised density domains superimposed on top. An overview of all the final density
of all bodies in the model required to fit the gravity field can be found in Table 1.

Both sedimentary layers of the model in Fig. 4 reflect trends across the region previously identified in the Fig. 3 cross sections,
with thicker and denser sediments in the Po Basin than in the Molasse Basin, and large thicknesses of consolidated sediments
in the central Alps (18 km) representing the Alpine nappe stacks. Maximum thicknesses of 9 km and 12 km were modelled in
the Po Basin for unconsolidated and consolidated sediments respectively, whilst 6 km and 9 km were modelled in the Molasse
Basin. Thicknesses of 3.75 km unconsolidated sediments were modelled in the deepest part of the Upper Rhine Graben with
consolidated sediments of up to 3 km. In both of the modelled sedimentary layers separate density domains were necessary in
the Eastern Molasse Basin (2470 kg/m3 and 2680 kg/m3) and the Po Basin (2470 kg/m3 and 2700 kg/m3) that were denser
than the sediments in the rest of the region (2450 kg/m3 and 2670 kg/m3).

The modelled configuration of the European upper crust is thicker but a similar density on average (20 km and 2700 kg/m3),
compared to the Adriatic upper crust (12 km and 2700 kg/m3). The thickest regions of upper crust can be found around the
Bohemian massif in the northern foreland and the Brianconnais Terrain and Tauern Window in the Alps at up to 30 km thick,
whilst thinned upper crust is modelled below the Adriatic Sea and the Ivrea Zone at only 4 km thick. Multiple density domains
in the upper crust correspond to known tectonic features in the modelled region such as the Variscan domains of Saxothuringia
(2670 kg/m3) and Moldanubia (2700 kg/m3), the massifs of Bohemia (2740     kg/m3) and Vosges/Black Forest (2660
kg/m3) that lie close enough together in the model to be grouped, the Ivrea Body (2790 kg/m3) and the Apennine plate (2720
kg/m3). However, in the Alps and the Adriatic Sea the density domain boundaries modelled were found not to correspond to
specific tectonic features.  The Alps are divided roughly North East (2740 kg/m/3) to South West (2670 kg/m3), denser in the
NE and the Adriatic Sea is split roughly North (2660 kg/m3) to South (2700 kg/m3), denser in the south.


The modelled configuration of the European lower crust is of similar thickness, but less dense on average (10 km and 2860 kg/m3) compared to the Adriatic (10 km and 2910 kg/m3). It was found necessary to model the lower crustal Alpine root thicker and denser (2950 kg/m3 and 34 km) than the rest of the region. Density domains within the lower crust correspond less with known tectonic features than those in the upper crust. Of the domains in the lower crust only two correspond roughly to tectonic features, one to the Saxothuringian Variscan domain (2920 kg/m3) and one to the Ivrea Body (3100 kg/m3). A large

region of similar density within the lower crust exists, mostly on the European Plate, covering an area including the Moldanubian Variscan domain, the Bohemian Massif and the Western and Eastern Alps (2800 kg/m3). The central Alps and the western Po Basin are also grouped as a region of similar density (2950 kg/m3). As with the upper crust, the lower crust beneath the Adriatic Sea is split roughly North (2750 kg/m3) to South (3040 kg/m3) with a denser domain in the south. Lower crustal densities on the European and Adriatic plates have been modelled as low as 2800 kg/m3 and 2750 kg/m3 respectively

and although necessary to resolve the gravity anomaly, these values are more similar to upper crustal density values. The other density domains in the lower crust of the region have values that would be expected for this depth level.

Figure 5 show the depths to the surfaces of the Moho and LAB utilised in this work. The Moho can be seen to be shallowest below the Ligurian Sea (20 km) but also shallow at the Ivrea body (22.5 km) and below the Upper Rhine Graben (25 km). It reaches its deepest point in the crustal root of the Alps at 55 km. From the gravity modelling process it was found necessary

to have variation in the density of the lithospheric mantle and that the regions of different density correspond to different thicknesses of the lithosphere (Geissler, 2010). Broadly the lithosphere is thinnest and least dense in the North-West of the region (70 km and 3305 kg/m3) whilst being thickest and densest in the South-East (140 km and 3335 kg/m3) below the Adriatic Sea. The shallowing of the LAB below the Alps can be seen to correspond to the boundary between the Austro Alpine and Helvetic/Penninic Alps.

The full extent of the observed gravity of the modelled region is visible in Fig. 2 and the calculated gravity response is visible in Fig. 6. The residual anomaly of the generated model can also be seen in Fig. 6 demonstrating the close match achieved by the generated structural model. Almost all of the modelled area reproduces the observed gravity to ± 25 mGal with the exception of a couple of isolated regions where the misfit between observed and calculated gravity slightly exceeds that.

The thickness and average density of the modelled crust throughout the region are shown in Fig. 7. The thickness of the entire

crust is calculated as the difference between the top surface of the upper crust and the top surface of the lithospheric mantle from the model. The lateral variation in average density is calculated as a weighted average taken from the thicknesses and densities of the upper crust and the lower crust at every point in the model.

Overall the crust can be seen to be thicker and less dense on average on the European plate (27.5 km and 2750 kg/m3) compared to the Adriatic (22.5 km and 2850 kg/m3). The thickest crust is found in the crustal root of the Central Alps at around 55 km

thick. Areas of thinned crust are found below the sedimentary depocentres of the Po Basin and the Upper Rhine Graben, which can additionally be seen extending South and West of its surface location, however the crust does not appear significantly thinned beneath the Molasse Basin. Whilst the Adriatic crust is denser on average than the European crust it has more extreme density variations within it, such as a modelled low density crust in the North of the Adriatic indenter that coincides with the



Veneto-Friuli plain (2700 kg/m3), immediately adjacent to much denser crust lying to the South below the Adriatic Sea (2900
kg/m3).

Density contrasts within the crust correlate spatially with the locations of some Alpine zone boundaries as defined in the
literature (Schmid et al., 1989; Schmid et al., 2004). The Brianconnais terrain can be seen as a higher density block contrasting
with the rest of the zones that surround it and the Southern Alps can also be seen as a dense block, with its borders to the
Brianconnais terrain and the Austro-Alpine zone clearly defined in the East of the modelled region. The Tauern window can
also be seen clearly as a low density anomaly within the Austro-Alpine zone.

Figure 8 also shows the thickness and average density of the modelled crust, but additionally shows their correspondence with
seismicity and deformation. The thickness of the crust is overlain with present day vertical displacement rates (Sternai et al.,
2019) in Fig. 8a and the average density of the crust is overlain with present day horizontal surface strain distribution (Sánchez
et al., 2018), seismic events of a moment magnitude of 6 or larger (Fäh et al., 2011; Stucchi et al., 2012; Grünthal et al., 2013)
and the location of modelled upper crust domain boundaries in Fig. 8b, so that relationships between crustal features and
deformation and seismicity can be interrogated.

Within the Alps a strong correlation exists between the thickness of the crust and vertical displacement rates at the surface.
Regions of modelled thickened crust correspond to high positive rates of vertical displacement, such as within the Alps.
Regions of thinned crust, such as in the Po Basin and the Upper Rhine Graben, were found to correspond to negative vertical
displacements. Differing rates of vertical displacement can also be observed in the Western Molasse and Eastern Molasse
Basin, with the west uplifting and the east subsiding. The transition between these two behaviours in the Molasse Basin
corresponds to the boundary of the modelled density domain boundaries in the sedimentary and upper crustal layers, separating
the denser eastern region of the basin and the less dense western portion of the basin.

All seismic events and surface strain distribution displayed in Fig. 8b correspond to changes in density of the crust in the
model. Regions with a pronounced change in the density of the crust such as at the edges of the Brianconnais terrain and low
density block in the north of the Adriatic Indenter are coincident with seismic events. Whilst not every seismic event
corresponds to a contrast in the average density of the crust, they all correspond to the location of density domain boundaries
within the upper crust as defined in the generated model. Horizontal surface strain distribution also corresponds to the location
of density domain boundaries within the upper crust as defined in the model, with the direction of maximum horizontal strain
predominantly perpendicular to the domain boundaries.

**5 Discussion**

Differing methods of classifying the Alpine zones have been adopted over time, however the results presented here would
support works that utilise tectonic reconstructions and constrain zones based on the plate the crust originated from (e.g. Schmid
et al., 2004). Crust derived from different terrains could potentially be assumed to have differing properties such as density,
and from the model produced in this work that is found to be the case. From the results, correlation can be observed between
zones of different density in the model and Alpine zones as defined by tectonic reconstructions and paleogeography, such as



the dense Brianconais Terrain and Southern Alps and the less dense Tauern Window. As no density domain geometries were pre-defined during the modelling stage, the correlation of these domains within the generated model to known features adds validity to the generated model.

Additionally, Alpine zones of Adriatic provenance were found in general to be denser and those of European provenance to be less dense, a trend also noted in the present day densities of the Adriatic and European crusts, potentially indicating that prior to orogenesis this was also the case. Adriatic continental crust derived Alpine zones such as the Austro-Alpine and Southern Alps appear denser in general than the European continental crust derived Helvetic zone and Tauern window. The Brianconais terrain derives from neither Europe nor Adria and as such appears in the results of this work, distinct from both,

as the region of highest density in the modelled area. These observations are consistent with the interpretation that the provenance of crust within the Alps can potentially be indicated by its properties, such as density, implying that as the Alpine zones were emplaced at different times during orogenesis, the respective plates prior to orogenesis must have had similar crustal properties to the present day.

Regions in the generated model exist with similar provenance and differing densities, indicating that factors other than the

provenance of zones also influence their densities. This is exemplified by the Helvetic and Penninic Alps, both deriving from the European plate, possessing a boundary between them that is clearly visible as an average crustal density contrast. Additionally, some expected boundaries between crusts of different provenances within the generated model are obscured by other elements of the model. The transition from the European, Helvetic and Peninnic, to Adriatic Austro Alpine units corresponds to the thickest area of the crustal root, where lower crust percentages are much higher than upper crust, creating

a region of high density crust in the model that masks the transition from European to Adriatic crust when looking at the average densities of the crust.

Correlation between present day horizontal surface deformation and large seismic events with density contrasts within the crust of the generated model would suggest the localisation of deformation along these features. Previous works have also shown correspondence between the localisaton of seismicity at density contrasts within the crust such as at crustal block

boundaries (Dentith and Featherstone, 2003), providing further validity to the model generated in this work.

All large seismic events in the region correspond to the modelled location of upper crustal density domain boundaries, however they do not all correlate to contrasts in the average density of the crust, potentially indicating that within the Alps upper crustal density contrasts are a more likely location for the localisation of seismicity than in the lower crust. Observations of the occurrence of seismicity at depth within the Alps have shown that it is predominantly present within the upper crust (Wiemer,

et al., 2017), supporting interpretations made from the derived model. However, regions exist within the model that have both an average crustal density contrast and an upper crustal density domain boundary that have no seismic events, indicating that there are additional controlling factors to the localization of seismicity.

Observations of the correlation between positive vertical displacement at the surface (Sternai et al., 2019) and thickened crust within the modelled region, and negative vertical displacement and thinned crust also strengthen the validity of the model,

with his behaviour expected due to isostasy. From the thickness of the crust it can also be seen that it is significantly thinned





beneath the Po Basin of the southern foreland and not in the Molasse Basin of the northern foreland, explaining the discrepancy in deposited sediment thicknesses noted prior. This could also indicate different driving mechanisms for the formation of either basin, with the Molasse Basin potentially lacking significant subsidence due to extension, forming predominantly through flexural means and the Po Basin forming through both flexural and active extensional processes.

The results presented in this work indicate crustal properties that would support observations from previous works into the dynamics of the region. Whilst correlation can be observed between vertical displacement at the surface and thickened or thinned crust, some regions such as the Molasse Basin feature a crust of similar thickness throughout but also feature a change in the polarity of surface vertical motion. The crustal densities of the model generated here would support this change however, with the transition occurring at the boundary of a density domains in the crust and the denser eastern portion exhibiting

subsidence and the less dense western portion exhibiting uplift.

Previous works into the dynamics of the Adriatic plate, show that it acts as a more rigid indenter than the European plate as it moves northwards rotating counter-clockwise into it (Nocquet and Calais, 2004; Vrabec and Fodor, 2006). Results of the generated model show the Adriatic crust is denser than the European crust and seismic velocities are also observed to be higher in the Adriatic crust than the European crust (IESG, 1978; IESG & ETH Zuerich, 1981; Strößenreuther 1982; Mechie et al

1983; Zucca 1984; Gajewski & Prodehl 1985; Deichmann et al, 1986; Gajewski et al 1987; Gajewski & Prodehl 1987; Yan and Mechie 1989; Zeis et al 1990; Aichroth et al 1992; Guterch et al 1994; Ye et al 1995; Scarascia and Cassinis, 1997; Enderle et al 1998; Bleibinhaus & Gebrande, 2006; Brueckl et al., 2007). Higher densities and velocities indicate a more mafic lithology for these domains potentially suggesting that it may be stronger than the European crust. The properties of the plates as modelled here would suggest that in the converging Eastern Alps the denser Adriatic crust would subduct under the European

crust which fits with the subduction polarity identified from teleseismic tomographic work in the region (Lippitsch et al., 2003) and high-resolution receiver function analysis (Hetényi et al. 2018).

Although correlations are noted between lateral variation in density distribution in the crust and observations such as plate dynamics and the localization of seismicity and deformation, the causes of these observations can not necessarily be constrained. Planned future modelling work will look closer at the features of the crust by creating thermal and rheological

models to give further explanation as to the driving forces behind the observed correlations, potentially helping to better explain trends noted in this work. Work is also progressing on constraining deeper structures in the region, such as the mantle, allowing for better constraints on crustal features in the future.

Although the generated model fits the observed gravity well across almost all of the modelled region, it represents a simplified version of the geology below the surface that is not able to account for all the complexity of the real world equivalent and as

such, inaccuracies within the generated model exist. Additionally, whilst the location of density domains in the generated model remains a non-unique solution, efforts were made to minimise errors by using seismic data and indications from previous modelling work to constrain the densities within each layer and density domain boundaries. Whilst these uncertainties cannot fully be accounted for, by only dealing with features and trends appropriate to the scale of the modelling work they are severely





mitigated. At the scale of the Alps and their forelands as described in this work, irrespective of localised changes to surface
heights or densities, the overall trends identified in this work would not be altered.

**6 Conclusion**

By creating the first gravity constrained, 3D, structural and density model of the lithosphere focused on the Alps and their
respective forelands, insights were gained into the distribution of densities at depth within the crust. The findings suggest that
the present day Adriatic crust is both thinner (22.5 km) and denser (2800 kg/m3) than the European crust (27.5km, 2750
kg/m3). Crust derived from different terrains was also found to have significantly different densities with Alpine zones of
Adriatic provenance. The Austro-Alpine and Southern Alps were found to be denser and those of European provenance such
as the Helvetic Zone and Tauern Window to be less dense, indicating the respective plates prior to orogenesis may be assumed
to have had similar crustal properties to the present day.

The model generated reproduced a good fit to the observed gravity field in the region with maximum misfits of around ± 25
mgal across the whole region and was further validated by density domains defined in the model corresponding to known
tectonic features, seismicity corresponding to crustal density contrasts and surface vertical displacements corresponding to
crustal thicknesses. The causes of these observations and correlations are unable to be explained solely from the results of this
work, therefore planned future modelling work will generate thermal and rheological models to give further insight into the
crustal architecture of the region as well the causes of the localization of deformation and seismicity within the region.


**Author Contributions**

Hans-Jürgen Götze and Jörg Ebbing contributed pre-existing gravity models from the region and advised on the gravity
modelling workflow and utilisation of the software used to carry it out. György Hetényi contributed seismic data for the work
and advised on the interpretation of the crust and Moho in the Eastern Alps. Magdalena Scheck-Wenderoth advised on the
entire workflow and the interpretation of results. Cameron Spooner carried out the modelling work and prepared the manuscript
with contributions from all co-authors.

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

- Figure 3. Cross sections through generated model showing thickness of model layers. Location is marked in Fig. 1a. Lithospheric mantle layer is shown in red, lower crust is shown in grey, upper crust is shown in brown, consolidated sediments are shown in blue and unconsolidated sediments are shown in yellow. In Fig 3. a) Density domains within each layer are shown as a change of shade and the densities of each domain are labelled. The observed and calculated response of the gravity throughout the cross section are shown. Fig. b) is scaled to show the thicknesses of all layers down to the aesthenospheric mantle, shown in orange.

- Figure 4. Thickness of a) unconsolidated sediments, b) consolidated sediments, c) the upper crust and d) the lower crust across the modelled area. Density domains required within the layer are overlain in white, layer numbers are shown in white and correspond to Table 1. Locations of key tectonic features are overlain (abbreviations shown in Fig. 1a caption).

- Figure 5. a) Depth to top surface of the Moho across the modelled area. Density domains required within the layer are overlain in white, layer numbers are shown in white and correspond to Table 1. b) Depth to top surface of the LAB from Geissler (2010) across the modelled area. Solid lines demark the boundaries of Alpine zones, the dotted black lines indicate the extent of the unaccreted Adriatic plate. Locations of key tectonic features are overlain for both figures (abbreviations shown in Fig. 1a caption).

- Figure 6. a) Calculated gravity of the region at 6km above the datum resulting from the final structural and density model produced in this work. b) Residual gravity after calculated gravity has been subtracted from observed gravity in the region. Solid lines demark the boundaries of Alpine zones, dotted black lines indicate the extent of the unaccreted Adriatic plate. Locations of key tectonic features are overlain for both figures (abbreviations shown in Fig. 1a caption).

- Figure 7. a) Thickness and b) average density of the entire crust across the modelled area. Solid lines demark the boundaries of Alpine zones, the dotted black lines indicate the extent of the unaccreted Adriatic plate. Locations of key tectonic features are overlain (abbreviations shown in Fig. 1a caption).

- Figure 8. a) Thickness of the crust across the modelled area overlain with vertical displacement rates (Sternai et al., 2019). Dotted black lines indicate isolines relating to the vertical displacement rates in mm/a. Regions where the overlain data was not available have been whited out. Locations of key tectonic features are overlain (abbreviations shown in Fig. 1a caption). b) Average density of the crust across the modelled area overlain with geodetically derived horizontal surface strain distribution from (Sánchez et al., 2018) and seismic events of a Moment Magnitude over magnitude 6 (Fäh et al., 2011; Stucchi et al., 2012; Grünthal et al., 2013). Stick orientation indicates orientation of maximum surface strain. Dotted black lines indicate the upper crust density domains of the final structural and density model produced in this work. Regions where the overlain data was not available have been whited out. Locations of key tectonic features are overlain (abbreviations shown in Fig. 1a caption).

**Tables**



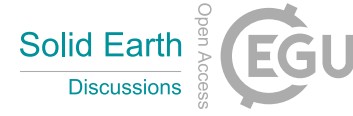

Table 1 -

| Unit | Mean Density indicated by P Wave Velocities (Kg/m3) | Density Used in Final Model (Kg/m3) |
|---|---|---|
| 1. Unconsolidated Sediments | 2530 | 2450 |
| 2. Unconsolidated Sediments – East Molasse | 2540 | 2470 |
| 3. Unconsolidated Sediments - Po | 2610 | 2470 |
| 4. Consolidated Sediments | 2680 | 2670 |
| 5. Consolidated Sediments – East Molasse | 2670 | 2680 |
| 6. Consolidated Sediments - Po | 2700 | 2700 |
| 7. Upper Crust – Saxothuringia | 2690 | 2670 |
| 8. Upper Crust – Moldanubia | 2710 | 2700 |
| 9. Upper Crust – Bohemia | 2720 | 2740 |
| 10. Upper Crust – Vosges and Black Forest | 2690 | 2660 |
| 11. Upper Crust – East Molasse | 2720 | 2720 |
| 12. Upper Crust – East Alps | 2740 | 2740 |
| 13. Upper Crust – West Alps | 2740 | 2670 |
| 14. Upper Crust – Po | 2740 | 2730 |
| 15. Upper Crust – North East Adria | 2780 | 2660 |



| 16. Upper Crust – Ivrea | - | 2790 |
| 17. Upper Crust – East Adria | 2780 | 2700 |
| 18. Upper Crust – Apennine | 2770 | 2720 |
| 19. Lower Crust – Saxothuringia | 2900 | 2920 |
| 20. Lower Crust – Europe | 2890 | 2800 |
| 21. Lower Crust – Alps | 2880 | 2950 |
| 22. Lower Crust – Ivrea | - | 3100 |
| 23. Lower Crust – Northern Adria | 2990 | 2750 |
| 24. Lower Crust – East Adria | 2950 | 3040 |
| 25. Lithospheric Mantle – Less Dense | 3340 | 3305 |
| 26. Lithospheric Mantle – More Dense | 3260 | 3335 |
| | | |
| Water | - | 1025 |
| Asthenospheric Mantle | - | 3320 |

530

**Table Captions**

- Table 1. The density of each domain in the model indicated by converting from its mean P wave velocity using the empirical relationship detailed in Brocher (2005) and the density of all domains used in the final model of the region found to best reproduce the
535 indications of both the seismic data sources and the gravity field. Locations of each density domain can be found in Fig. 4.



Figure 1

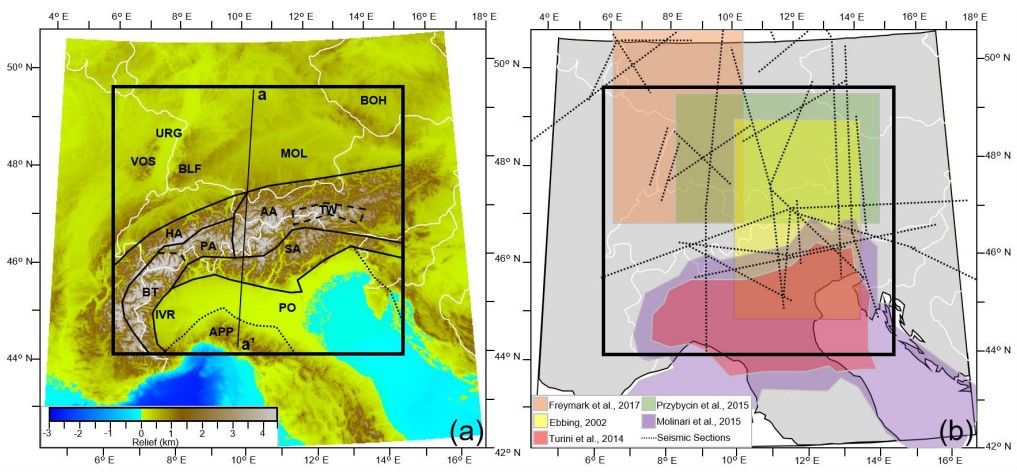



Figure 2

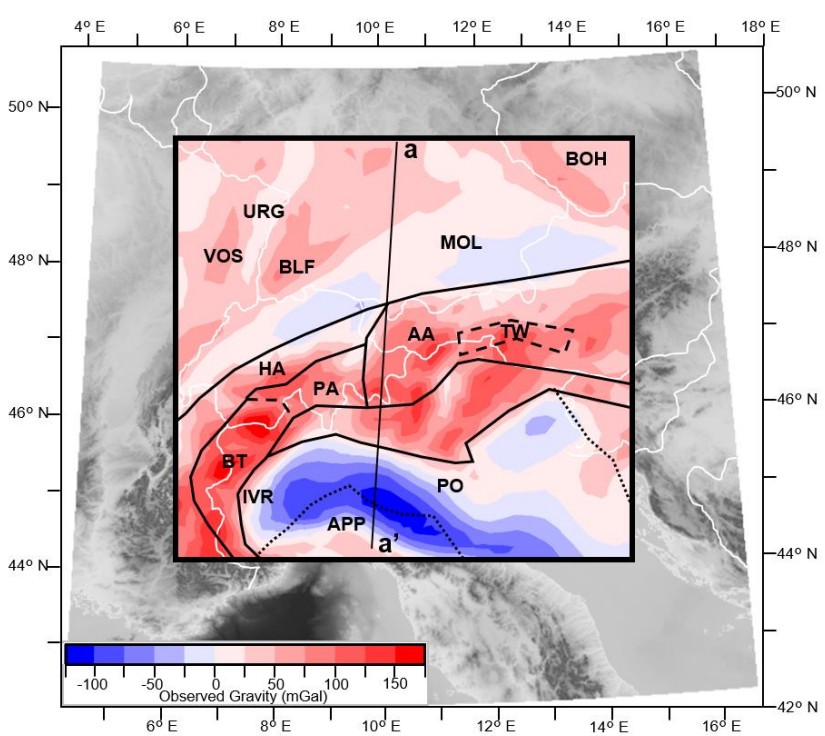



Figure 3

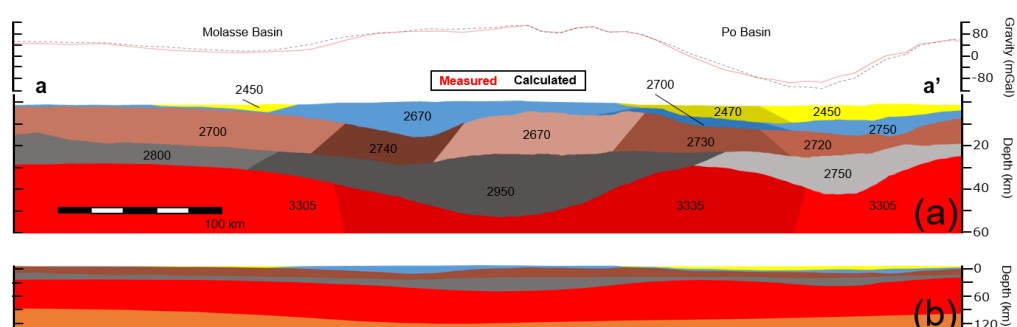




Figure 4

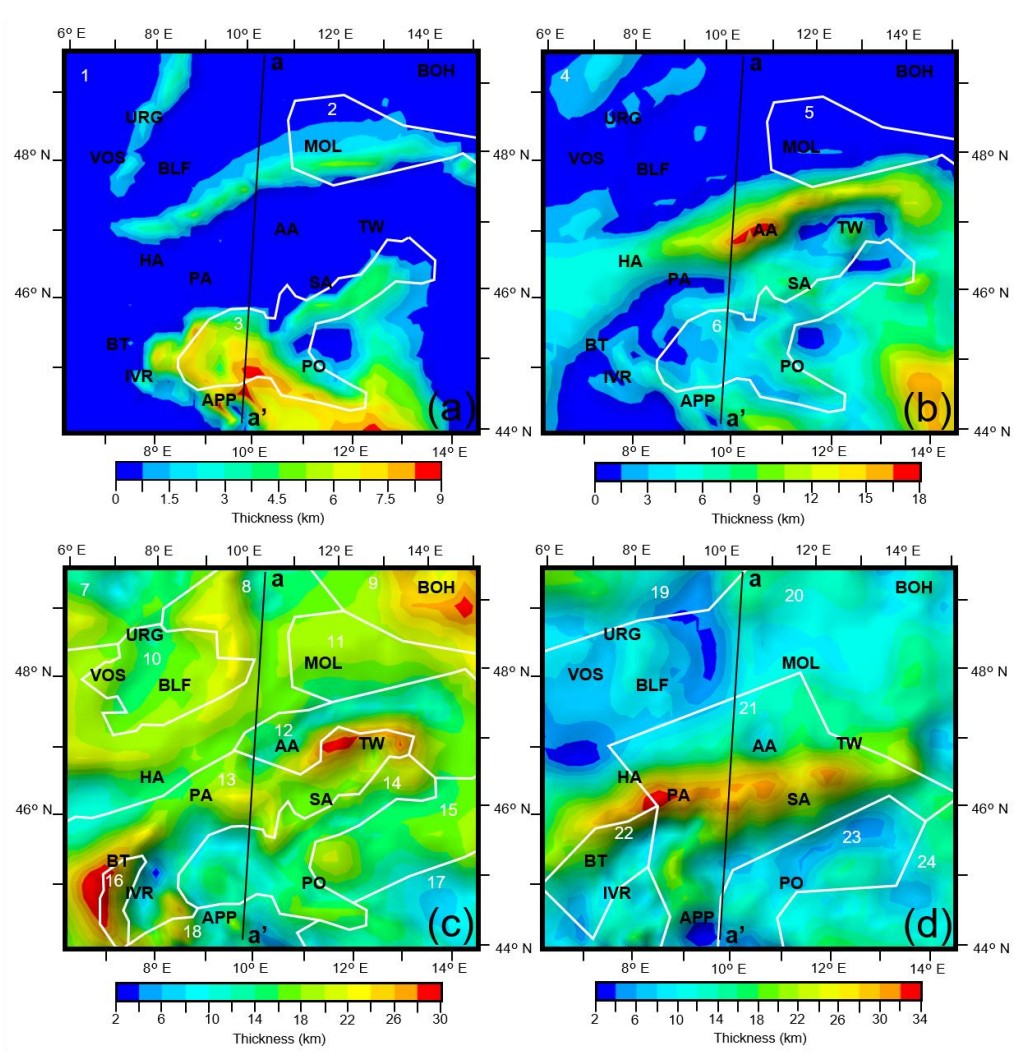



Figure 5

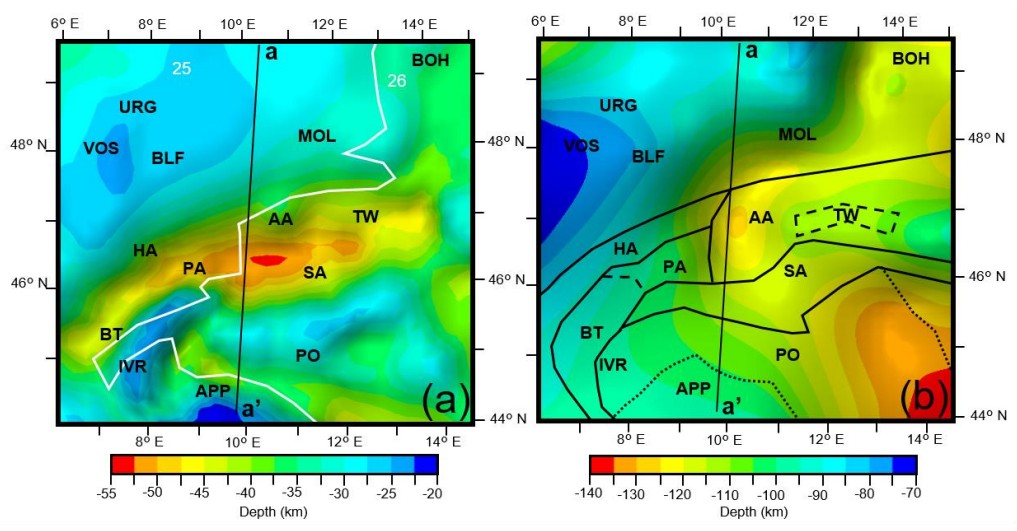





Figure 6

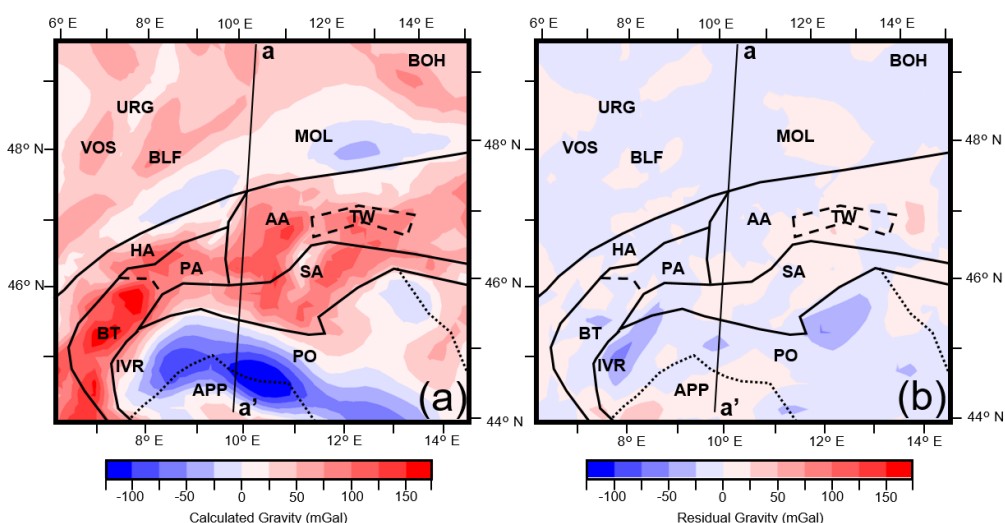



Figure 7

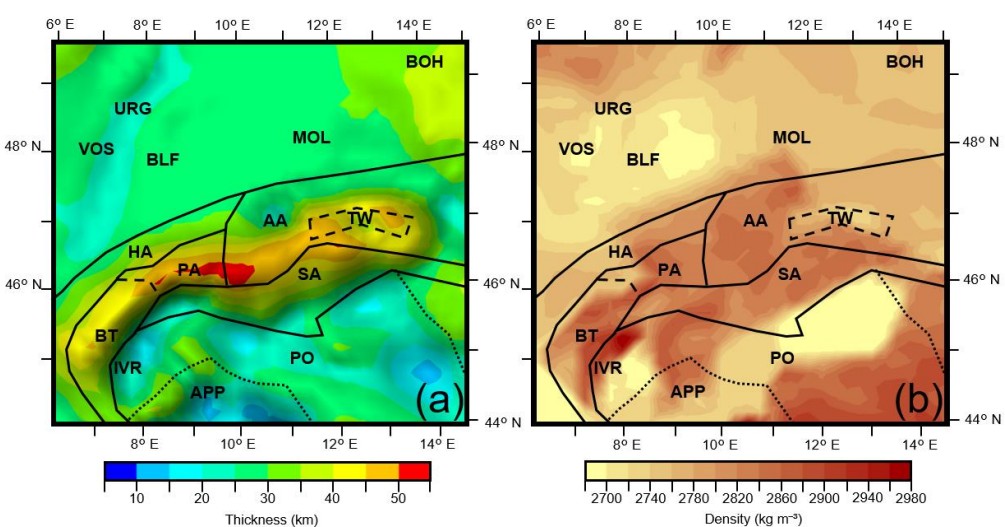





Figure 8

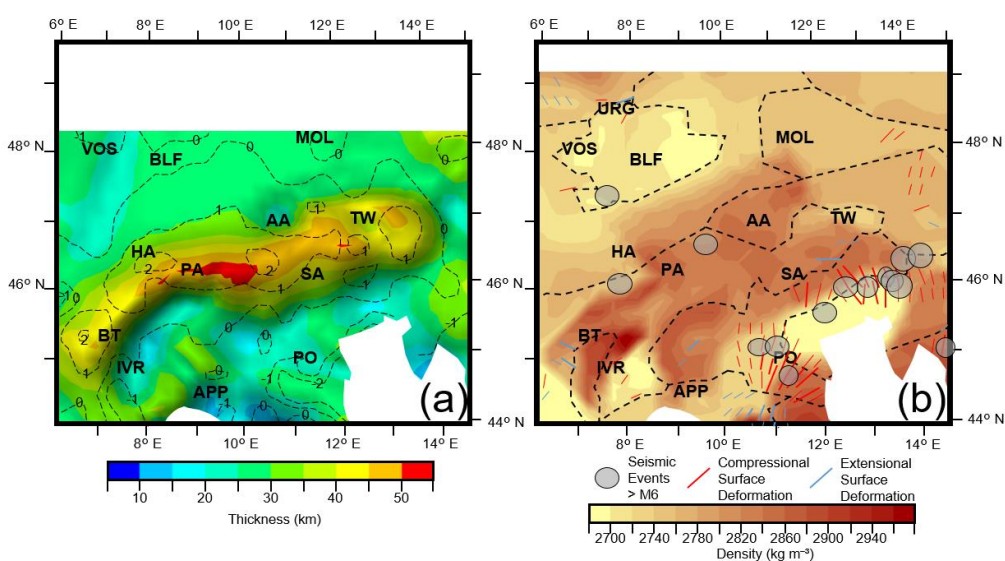