# Peer review of "Density distribution across the Alpine lithosphere constrained by 3D gravity modelling and relation to seismicity and deformation"

_Solid Earth, 2019_

## Referee Comment (RC1) · Anonymous Referee #1 · 12 Aug 2019

This paper presents a novel seismically constrained, 3D, structural and density model of the lithosphere of the Alps and their respective forelands. It was constructed through the integration of numerous geoscientific datasets, and was adjusted to fit the observed gravity field. The obtained results show that significant more heterogeneity is required in the crust than in other model layers to replicate the gravity field, and that notable differences exist between the crust of the European and Adriatic plates. Moreover, it is observed that the distribution of seismicity and deformation within the studied region strongly correlates with thickness and density changes within the crust. These results are novel and original, and constitute a great contribution to the understanding of the geological setting and geodynamic evolution of the studied area. The conclusions of this paper are of interest to the broad geoscience community, particularly for researchers focused on the investigation of Alpine crustal evolution and configuration. It is also very important to highlight that the model presented in this paper is available for open access, being possible to use it to further discuss and investigate crustal provenance and crustal properties variations across the studied region.

However, I have a couple of concerns that surely can be properly addressed by the authors:

1) It would be worthy to provide some kind of sensitivity analysis about the impact of subtle changes in density and/or density domains boundaries geometries on the gravity response of the model. Such sensitivity analysis could further support and reinforce your interpretations and conclusions and contribute to reduce the inherent ambiguity of the gravity method. 2) You should discuss the causes and/or possible origin of the differences observed between the densities you calculated for each modelled body from P wave velocities and the densities you finally used, which are shown in Table 1. Particularly when considerable differences exist (e.g. Upper Crust North East Adria, Lower Crust Northern Adria) and when opposite tendencies arose (e.g. Lithospheric Mantle Less Dense vs. Lithospheric Mantle More Dense). 3) You should also provide possible explanations or at least discuss the fact that very low densities had to be assigned to lower crustal bodies (Lower Crust Europe, Lower Crust Northern Adria) corresponding to some sectors of the European and Adriatic plates, in order to fit observed and calculated gravity anomalies. Mostly, considering that such very low densities are more typical of the upper than of the lower crust. 4) Language should be polished. Some sentences are too long. Wording and phrasing should be improved.

Finally, other comments, suggestions, questions and corrections are indicated in the attached pdf file.

Best regards,

Please also note the supplement to this comment:
https://www.solid-earth-discuss.net/se-2019-115/se-2019-115-RC1-supplement.pdf

---

## Referee Comment (RC2) · Andrea Argnani (Referee) · 12 Aug 2019

General comments The paper presents an interesting attempt to produce a 3D density distribution in the Alpine lithosphere, constrained by gravity modelling and by a wealth of geophysical data, mostly taken from the literature. The modelling results support the occurrence of differences in crustal thickness and density between the European and Adriatic domains, including the units now stacked in the Alpine belt, which are inferred to maintain a specific density characterization that reflects their provenance. Partly comparable results showing differences in thickness and density between the European and Adriatic crust were presented in a previous study, although carried out

along a transect (TRANSALP; Ebbing et al., 2006, Tectonophysics) and not presenting a 3D picture. That work, however, should be cited at some stage. In spite of the uncertainties in precisely locating the density distribution using gravity modelling, the results are in many aspects convincing and stimulating when compared to the geological interpretation of the Alpine structure. The presentation of the results and the Discussion sections are very essential and could be expanded a little in order to compare the results with other works and to give some hints on the potential implications for the tectonic evolution of the Alps. Some issues that may be worth expanding/commenting further are listed in the Specific comments below.

Specific comments lines 47-49: modelling indicates that the crust is thinner in the Po and Molasse basins, where sedimentary depocenters are present, and in the Rhine Graben. Whereas crustal thinning occurred in the Rhine Graben in late-post Alpine timing, the Po and Molasse basins are produced by flexure, mainly due to the load of the adjacent mountain belts (Apennines and Alps, respectively). The thickness of the crust is not affected by plate flexure; the crustal thinning should therefore be considered an inherited feature. lines 207-208: perhaps some comments on the comparison of the results with previously published Moho maps (e.g., Ziegler and Dezes 2008, Geol. Soc. London; Spada et al., 2013 GJI) could be useful. lines 217-219: interestingly the two areas of negative residual anomaly in Fig. 6b seem to have some geological relationships. The western one partly follows the Ivrea zone, and the eastern one is over a basement high that existed since Late Permian-Triassic (see Masetti et al., 2012 AAPG). Perhaps the authors have some comments on this fact, that doesn't look random. lines 231-235: the average density of the crust result from the integration of a column of rocks where different units are stacked one on top of the other; the domains cropping out at surface not always continue at depth, as illustrated in many of the geological cross sections across the Alps (e.g., Schmid et al 2004). The relationship between average density distribution and tectonic domains may not always be straightforward, and this should betaken into account, lines 246-248: the Molasse foreland basin originated by flexure of the European plate under the Alpine load, and that was

independent from the inherited along-strike difference in crustal density. Present-day vertical motions could represent a post-orogenic isostatic adjustment, but the Austrian sector of the Molasse is an undeformed basin, whereas a detachment connected to a thrust front located north of the basin underlies the western Molasse; an active involvement of the basement has also been suggested in the region of the Jura mountains (e.g., Mock and Herwegh 2017 Tectonics), and this could contribute to positive vertical motion. The authors also mention a difference in the density of sediments between the western and eastern Molasse basin; however, the difference in density between the two sedimentary domains is rather small: is it enough to drive a differential vertical motion? line 250-251: the distribution of earthquakes with M>6 and max. horizontal strain is rather limited spatially. They are mostly located at the thrust front of the Southern Alps which represents the active southern boundary of the Alps, as also supported by focal mechanisms (see Serpelloni et al., 2006). The difference in average crustal density observed in the model is expected at such plate boundary. The other few and sparse earthquakes are not very indicative of dynamics at crustal boundaries and likely reflect different tectonic regimes: the earthquake next to BLF is likely related to the Rhine Graben, whereas those in the Swiss Alps reflect a regional trend of extension/strike-slip that characterizes the highest regions of the central Alps, irrespective of tectonic domains. lines 282-285: the "boundaries" between different crustal "blocks" is a likely place for the occurrence intraplate earthquakes, that tend to follow pre-existing weakness zones. In the presence of an active plate boundary, like the Alps, the link between different blocks and seismicity is less obvious, as there are plate interfaces, and faults originated by the collisional process are abundant, and often seismically active (e.g., Serpelloni et al., 2016). Lines 289-290: see also Serpelloni et al 2016 Tectonophysics for distribution of seismicity in central-eastern Alps: crustal seismicity seems to follow the major faults driving the eastward escape of the Eastern Alps . lines 295-300: as mentioned before the thickness of the crust underlying the Molasse and Po foreland basins should be taken as unaffected by the load of the mountain belts. The possibility of having a contribution to subsidence driven by crustal extension in the Po Basin,

as suggested by the authors, seems highly unlikely and no evidence to support it is present. Lines 306-307: see also Serpelloni et al 2016 Tectonophysics for distribution of horizontal velocity and strain rates in central-eastern Alps: the motion of Adria seems mostly accommodated by deformation at the thrust front of the Southern Alps.. line 312-316: the authors assume that the evidence for a thinner and denser crust in eastern Adria supports its subduction underneath the European plate, as originally inferred by Lippitsch et al. However, it should be considered that the long term evolution of the Alps, including the Eastern Alps, is consistent with a subduction of the the European plate below Adria. And this is certainly true until the last 20 Myr. Therefore, density alone does not justify a supposed change in the polarity of subduction along the strike of the Alpine orogen. Moreover, shortening of the Adriatic plate in the eastern Southern Alps is rather limited, as also pointed out by Kastle et al. (2019 SE), and is not enough to explain the extent of the slab observed in Lippitsch's tomography.

Figure 2. It would be useful to have also a simplified geological map of the Alps (e.g., taken from Schmid et al 2004) to give a better link between geophysical and geological data. Figure 3: A simplified geological cross section, plotted at the same scale of the profile in Fig. 3, would be useful to give a better feeling of the relationships between density domains and geological units. I am not aware of a geological cross section running along the same direction of the profile in Fig. 3, but perhaps the TRANSALP cross section, with appropriate comments, could be indicative enough (after Pfiffner 2014, Geology of the Aps; or Schmid et al 2004)

Technical corrections line 13: "orogenies" instead of "orogenys" line 43: "More recent" instead of "Newer" line 79: sufficient; however (insert semicolon) Line 127: "before" instead of "prior" line 185: "... thicker, but with a similar..." line 191: "Apennine belt" instead of "Apennine plate" line 204: "respectively, " instead of "respectively" (insert comma) line 218: "exceeds that value." instead of "exceeds that." line 297: "before" instead of "prior" Line 303: "however" can be removed line 489: before listing the labels of key tectonic features add that a-a' is the cross section in Fig. 3. line 499:

add that a-a' is the cross section in Fig. 1a line 500: location is marked in Figs 1a, 2 and 4 to 6. line 508: "depth to the Moho" instead of "depth to top surface of the Moho" line 508: "... required within the lithospheric mantle..." instead of "...required within the layer..."

―――――――――――――――――――

---

## Author Comment (AC1) · 1 Oct 2019

Reviewer 1

- It would be worthy to provide some kind of sensitivity analysis about the impact of subtle changes in density and/or density domains boundaries geometries on the gravity response of the model. Such sensitivity analysis could further support and reinforce your interpretations and conclusions and contribute to reduce the inherent ambiguity of the gravity method.

- You should discuss the causes and/or possible origin of the differences observed between the densities you calculated for each modelled body from P wave velocities and the densities you finally used, which are shown in Table 1. Particularly when considerable differences exist (e.g. Upper Crust North East Adria, Lower Crust Northern Adria) and when opposite tendencies arose (e.g. Lithospheric Mantle Less Dense vs. Lithospheric Mantle More Dense).

- You should also provide possible explanations or at least discuss the fact that very low densities had to be assigned to lower crustal bodies (Lower Crust Europe, Lower Crust Northern Adria) corresponding to some sectors of the European and Adriatic plates, in order to fit observed and calculated gravity anomalies. Mostly, considering that such very low densities are more typical of the upper than of the lower crust.

We thank the reviewer for these suggestions and have added an additional figure to the manuscript and some new paragraphs to address these topics collectively. The new Fig. 9, shows the results of a model that has been run with lower crustal densities indicated from P-wave velocity to density conversions in regions where the misfit between the densities indicated from P-wave velocity and those used in our final model are greatest (Europe and North Adria). The effect that these alterations have on the calculated and residual gravity fields of the model is then addressed in the paragraphs from line 343 – 363. Model sensitivity, the origin of differences between P-wave velocity to density values and modelled values, and causes for the regions of low density lower crust are also discussed.

- Language should be polished. Some sentences are too long. Wording and phrasing should be improved.

We thank the reviewer for the extensively annotated copy of the manuscript that they provided and have implemented all the corrections they have made to the text along with all suggestions on improvement of sentence structure and brevity. Additionally we have also made our own changes to portions of the text in an attempt to further deal with the point raised.

- Please, show in Figure 1a or in a new figure the limits/boundaries of showing the different plates, blocks and terrains, particularly showing the limits of the European and Adriatic plates, the Vosges, Black Forest and Bohemian massifs, the Po and Molasse Basins, the Upper Rhine Graben, the Veneto-Friuli plain, the Ivrea Zone, and Moldanubia and Saxothuringia. Where is the Ligurian Sea located? You should refer to a figure in the Introduction showing the terrains, blocks, plates mentioned in Introduction section.

We thank the reviewer for their helpful suggestions and have altered Figure 1 significantly to incorporate all of the limits and boundaries you have suggested. In addition the labelling of important tectonic features on all other figures has also been edited to include your suggestions. A sentence has also been added to line 38 of the manuscript indicating that the all tectonic features mentioned in the introduction can be seen in Fig. 1a. Figure captions have also been altered to reflect these changes.

- Please, show clearly in a map and in the section presented in figure 3 the location, extension and boundaries of the European and Adriatic plates and the location of Central Alps.

We thank the reviewer for suggesting how to add clarity to the figure and have implemented these changes.

- Line 73 - seismic reflection heights?..what do you mean with heights?..please clarify

We thank the reviewer for pointing out this ambiguous meaning. We have changes the word heights at line 76 to depths, as we are referring the depths below the surface of seismic reflections.

- Line 78 - Please, briefly describe how these authors obtained the LAB (which methodology did they use), as the LAB is one of the sufaces constituting your model.

We thank the author for pointing out this omission and have added that the LAB was obtained by S receiver functions of teleseismic events in the sentence at line 81.

- Line 172 - Do you mean "upper crust" or whole crust?

The sentence was referring to the whole crust and has now been changed to reflect as such at line 170. We thank the reviewer for noting this.

- Line 179-182 You should use a different word. The word "modelled" is confusing, as you previously said that you did not modify layers thicknesses (which were defined from constraining data) during gravity modelling with IGMAS+.
We thank the reviewer for pointing out the confusing wording and have updated the sentences at lines 178-181 to say the word 'used' instead of modelled.

-Line 185 – 188 Please, define European Upper crust, Adriatic Upper Crust and Adriatic Sea. Which bodies of the ones listed in Table 1 are supposed to compose them? Does Adriatic Sea correspond to bodies 15 and 17 in Table 1?
We thank the reviewer for pointing out the lack of clarity here and have made a number of changes to rectify this. The upper crust density domains that Europe (domains 7-11) and Adria (domains 14, 15 and 17) are composed of have been added at lines 184 and 185 respectively so that it is clear when viewing either Figure 4 or Table 1. The Adriatic Sea is now labelled in Figure 1a to make clearer where it is on the modelled area.

Line 195 – 196 Why are European and adreatic (Bodies 23 and 24?) average crust densities given different to those in table 1. Please explain.
We thank the reviewer for pointing out the lack of clarity here. The densities given are an average of the density domains that comprise it. This clarification has been added to the text at line 186 and we have added the lower crust density domains that Europe and Adria are composed of at lines 196 and 197 respectively so that it is clear when viewing either Figure 4 or Table 1.

Please, check along the whole paper the use of upper case or lowercase letters in the names of terrains, blocks etc. as Brianconais Terrain, Tauern Window, etc. Actually, you are sometimes using upper case and sometimes lowercase.
We thank the reviewer for bringing this to our attention and have checked the manuscript for mistakes such as this and corrected them all.

localisation or localization?. Please check spelling along the whole manuscript.
We thank the reviewer for noticing this and we have rectified the instances that these occurred.

What indicates the line labelled a-a´?. Please, explain it in Figure 1 and 2 caption.
We thank the review for pointing out this mistake. a-a' represents the cross section in Fig. 3. and we have updated the figure captions to reflect this.

---

## Author Comment (AC2) · 1 Oct 2019

Reviewer 2

- Partly comparable results showing differences in thickness and density between the European and Adriatic crust were presented in a previous study, although carried out along a transect (TRANSALP; Ebbing et al., 2006, Tectonophysics) and not presenting a 3D picture. That work, however, should be cited at some stage. We thank the reviewer for noting this omission and have now included it in the references and it is cited at line 59.

- The presentation of the results and the Discussion sections are very essential and could be expanded a little in order to compare the results with other works and to give some hints on the potential implications for the tectonic evolution of the Alps.

We thank the reviewer for mentioning this. We feel that as we have implemented many additional sections to both the Results and Discussion of the manuscript, in response to all issues raised during the review process, that this has been accomplished satisfactorily.

- lines 47-49: modelling indicates that the crust is thinner in the Po and Molasse basins, where sedimentary depocenters are present, and in the Rhine Graben. Whereas crustal thinning occurred in the Rhine Graben in late-post Alpine timing, the Po and Molasse basins are produced by flexure, mainly due to the load of the adjacent mountain belts (Apennines and Alps, respectively). The thickness of the crust is not affected by plate flexure; the crustal thinning should therefore be considered an inherited feature.

- lines 246-248: the Molasse foreland basin originated by flexure of the European plate under the Alpine load, and that was independent from the inherited along-strike difference in crustal density. Present-day vertical motions could represent a post-orogenic isostatic adjustment, but the Austrian sector of the Molasse is an undeformed basin, whereas a detachment connected to a thrust front located north of the basin underlies the western Molasse; an active involvement of the basement has also been suggested in the region of the Jura mountains (e.g., Mock and Herwegh 2017 Tectonics), and this could contribute to positive vertical motion. The authors also mention a difference in the density of sediments between the western and eastern Molasse basin; however, the difference in density between the two sedimentary domains is rather small: is it enough to drive a differential vertical motion? - lines 295-300: as mentioned before the thickness of the crust underlying the Molasse and Po foreland basins should be taken as unaffected by the load of the mountain belts. The possibility of having a contribution to subsidence driven by crustal extension in the Po Basin, as suggested by the authors, seems highly unlikely and no evidence to support it is present.

We thank the author for pointing out these mistakes and have made several changes to account for the issues raised. In the first instance we have changed the wording of the sentences at lines 47-50 such that references to crustal thinning have been removed, as this section purely deals with the geographical location of the sedimentary depocenters in the region. We agree with the reviewer that our work has not been able to constrain the underlying causes of features such as differential surface uplift in the East and West Molasse Basin and have added sentences reflecting this from lines 320-323. There we have also added mention to your suggestion that the thinned Po Basin crust could also be an inherited feature

- lines 207-208: perhaps some comments on the comparison of the results with previously published Moho maps (e.g., Ziegler and Dezes 2008, Geol. Soc. London; Spada et al., 2013 GJI) could be useful.

We thank the reviewer for pointing out this oversight. We have included a number of sentences from lines 211-216 commenting on the similarity with the trends of the Spada et al. (2013) Moho to our integrated Moho surface, whilst also mentioning why we believe our integrated Moho surface is better for the purpose of a 3D gravity model. We have also updated the references list accordingly.

- lines 217-219: interestingly the two areas of negative residual anomaly in Fig. 6b seem to have some geological relationships. The western one partly follows the Ivrea zone, and the eastern one is over a basement high that existed since Late Permian-Triassic (see Masetti et al., 2012 AAPG). Perhaps the authors have some comments on this fact, that doesn't look random.

We thank the reviewer for pointing out this omission. Indeed we had previously identified the source of these anomalies but neglected to include them in the manuscript. The anomalies are coming from Moho depths that are slightly too high, however as we have fixed surfaces during the modelling work and the anomaly was not coming from crustal densities we deemed it appropriate not to account for them in changes to density in the model. We have explained as such in additional sentences from lines 225- 230.

- lines 231-235: the average density of the crust result from the integration of a column of rocks where different units are stacked one on top of the other; the domains cropping out at surface not always continue at depth, as illustrated in many of the geological cross sections across the Alps (e.g., Schmid et al 2004). The relationship between average density distribution and tectonic domains may not always be straightforward, and this should betaken into account.

We thank the reviewer for their points here that we had not addressed the inaccuracies of constraining the horizontal boundaries of these features due to tectonic features at the surface not occupying the same location as their expression at depth. To address this we have added a new paragraph in the Discussion at lines 273-279, explaining that the density domains of the gravity model define a bulk density for different regions of the model and often correlate to tectonic features at the surface. Additionally, we now stress that caution should be exercised if trying to link the exact location of domain boundaries to features at the surface.

- lines 250-251: the distribution of earthquakes with M>6 and max. horizontal strain is rather limited spatially. They are mostly located at the thrust front of the Southern Alps which represents the active southern boundary of the Alps, as also supported by focal mechanisms (see Serpelloni et al., 2006). The difference in average crustal density observed in the model is expected at such plate boundary. The other few and sparse earthquakes are not very indicative of dynamics at crustal boundaries and likely reflect different tectonic regimes: the earthquake next to BLF is likely related to the Rhine Graben, whereas those in the Swiss Alps reflect a regional trend of extension/strikeslip that characterizes the highest regions of the central Alps, irrespective of tectonic domains.

- Lines 289-290: see also Serpelloni et al 2016 Tectonophysics for distribution of seismicity in central-eastern Alps: crustal seismicity seems to follow the major faults driving the eastward escape of the Eastern Alps .

- lines 282-285: the "boundaries" between different crustal "blocks" is a likely place for the occurrence intraplate earthquakes, that tend to follow pre-existing weakness zones. In the presence of an active plate boundary, like the Alps, the link between different blocks and seismicity is less obvious, as there are plate interfaces, and faults originated by the collisional process are abundant, and often seismically active (e.g., Serpelloni et al., 2016).

- Lines 306-307: see also Serpelloni et al 2016 Tectonophysics for distribution of horizontal velocity and strain rates in central-eastern Alps: the motion of Adria seems mostly accommodated by deformation at the thrust front of the Southern Alps.

We agree with the reviewer here on multiple points. That the large earthquakes away from the plate boundaries likely reflect different tectonic regimes and that the boundaries between different crustal blocks is a likely place for intraplate earthquakes. We have included Serpelloni et al. (2016) as a reference and added some additional points from that work to strengthen our discussion further. We added to the sentence at Line 259 to make it clearer that seismicity is primarily expected at major plate boundaries. And we also added multiple sentences to the paragraph from line 298, that also deals with why we used the sparse M>6 large earthquake dataset. As we only have the ability to describe density contrasts at a coarse resolution in the crust we felt it suitable. As mentioned before we do not have the ability to provide causal relationships from the correlation with earthquakes, hence we don't use a high resolution dataset. And we also lack the ability to provide causes for noted correlations as this will be interrogated in future works as is pointed out in the manuscript from line 364-369.

- line 312-316: the authors assume that the evidence for a thinner and denser crust in eastern Adria supports its subduction underneath the European plate, as originally inferred by Lippitsch et al. However, it should be considered that the long term evolution of the Alps, including the Eastern Alps, is consistent with a subduction of the the European plate below Adria. And this is certainly true until the last 20 Myr. Therefore, density alone does not justify a supposed change in the polarity of subduction along the strike of the Alpine orogen. Moreover, shortening of the Adriatic plate in the eastern Southern Alps is rather limited, as also pointed out by Kastle et al. (2019 SE), and is not enough to explain the extent of the slab observed in Lippitsch's tomography.

We thank the reviewers for pointing out the shortcomings of how we have addressed this topic in the text and have made changes to rectify accordingly, by adding additional sentences to the paragraph at line 339-342. We mention that there is no consistent plate model of the Alps, but that our findings in this work (bulk densities of the crust and lithospheric mantle of Europe and Adria) could help to reach this stage in the future.

- Figure 2. It would be useful to have also a simplified geological map of the Alps (e.g., taken from Schmid et al 2004) to give a better link between geophysical and geological data.

We thank the reviewer for this suggestion and have tried to implement accordingly. Overlays of additional tectonic features mentioned in the rest of the manuscript, such as the Ivrea zone, the Vosges, Black Forest and Bohemian Massifs have been added to the figure to complement the already labelled Alpine domains. We hope that the reviewer agrees this indicates a simplified version of the geological features in both the Alps and their forelands, and is an acceptable solution.

- Figure 3: A simplified geological cross section, plotted at the same scale of the profile in Fig. 3, would be useful to give a better feeling of the relationships between density domains and geological units. I am not aware of a geological cross section running along the same direction of the profile in Fig. 3, but perhaps the TRANSALP cross section, with appropriate comments, could be indicative enough (after Pfiffner 2014, Geology of the Aps; or Schmid et al 2004)

We thank the reviewer for this suggestion, but having looked into it we have decided against making any changes here. The location of the cross section used originally was chosen so that it represents most of the tectonic features within the region such as the European, Adriatic and Apennine plates, the Mollasse and Po basins and 2 different density domains within the Alps, thus giving a concise indicative view of our model, rather than multiple cross sections to achieve the same effect. As no geologic section exists of our chosen cross section we have been unable to implement a simplified geologic section next to it for reference. In an attempt to deal with this however we have labelled the extents of the European, Adriatic and Apennine plates in order to make interpretation of our cross section simpler.

- Technical corrections line 13: "orogenies" instead of "orogenys" line 43: "More recent" instead of "Newer" line 79: sufficient; however (insert semicolon) Line 127: "before" instead of "prior" line 185: "... thicker, but with a similar..." line 191: "Apennine belt" instead of "Apennine plate" line 204: "respectively, " instead of "respectively" (insert comma) line 218: "exceeds that value." instead of "exceeds that." line 297: "before" instead of "prior" Line 303: "however" can be removed line 489: before listing the labels of key tectonic features add that a-a' is the cross section in Fig. 3. line 499: add that a-a' is the cross section in Fig. 1a line 500: location is marked in Figs 1a, 2 and 4 to 6. line 508: "depth to the Moho" instead of "depth to top surface of the Moho" line 508: "... required within the lithospheric mantle..." instead of "...required within the layer..."

We thank the reviewer for noticing these mistakes and have implemented all of these changes into the manuscript.